# Impact of the COVID-19 Pandemic on the Therapeutic Continuity among Outpatients with Chronic Cardiovascular Therapies

**DOI:** 10.3390/ijerph191912101

**Published:** 2022-09-24

**Authors:** Manuela Casula, Federica Galimberti, Marica Iommi, Elena Olmastroni, Simona Rosa, Mattia Altini, Alberico L. Catapano, Elena Tragni, Elisabetta Poluzzi

**Affiliations:** 1Epidemiology and Preventive Pharmacology Service (SEFAP), Department of Pharmacological and Biomolecular Sciences, University of Milan, Via Balzaretti 9, 20133 Milan, Italy; 2IRCCS MultiMedica, Via Milanese 300, Sesto S. Giovanni, 20099 Milan, Italy; 3Department of Medical and Surgical Sciences-Pharmacology Unit, University of Bologna, Via Irnerio 48, 40126 Bologna, Italy; 4Department of Biomedical and Neuromotor Sciences-Hygiene and Biostatistics Unit, University of Bologna, Via San Giacomo, 40126 Bologna, Italy; 5Romagna Local Health Authority, Emilia-Romagna Region, Via A. De Gasperi 8, 48121 Ravenna, Italy

**Keywords:** COVID-19 pandemic, prescriptions, adherence, chronic treatments, cardiovascular diseases

## Abstract

The COVID-19 pandemic poses major challenges to healthcare systems. We aimed to investigate the impact of the pandemic on prescription and adherence patterns of chronic cardiovascular therapies (lipid-lowering [LL], oral antidiabetic drugs [AD], and antihypertensives [AH]) using administrative pharmaceutical databases. For each treatment, two cohorts of prevalent cases in 2019 and 2020 were compared. We evaluated the percentage change in dispensed packages and treatment adherence as a proportion of days covered (PDC). For all therapies, an increase was observed during March–April 2020 (LL: +4.52%; AD: +2.72%; AH: +1.09%), with a sharp decrease in May–June 2020 (LL: −8.40%; AD: −12.09%; AH: −10.54%) compared to 2019. The impact of the COVID-19 pandemic on chronic cardiovascular treatments appears negligible on adherence: 533,414 patients showed high adherence to LL (PDC ≥ 80%) in January–February 2020, and 2.29% became poorly adherent (PDC < 20%) in the following four-month period (vs. 1.98% in 2019). A similar increase was also observed for AH (1.25% with poor adherence in 2020 vs. 0.93% in 2019). For AD, the increase was restrained (1.55% with poor adherence in 2020 vs. 1.37% in 2019). The rush to supply drugs at the beginning of lockdown preserved the continuity of chronic cardiovascular therapies.

## 1. Introduction

The coronavirus disease 2019 (COVID-19) caused by severe acute respiratory syndrome coronavirus 2 (SARS-CoV-2) was first described in the city of Wuhan (China) in December 2019. On 30 January 2020, the World Health Organization (WHO) declared COVID-19 a Public Health Emergency of International Concern [1] and within a few months of its recognition, COVID-19 had reached more than 200 countries. The Italian government declared a state of emergency on 31 January 2020, introduced measures for social distancing on 23 February 2020, and enforced a complete lockdown of the country on 9 March 2020. Hospitals and Emergency Departments were forced to adjust rapidly to this completely new and rapidly evolving situation in order to manage an extraordinarily high number of contagious patients with respiratory symptoms [2,3,4].

However, in addition to the direct impact, the pandemic also determined a sudden and significant disruption in healthcare services for non-COVID-19 patients [5,6,7]. This may have had a dramatic effect on the quality of care and lives of persons of advanced age with complex chronic conditions who rely on uninterrupted access to medication [8,9,10]. 

As for all chronic diseases, the continuity of medication therapy is a cornerstone for the effective management of cardiovascular prevention [11,12]. Many factors can negatively influence the continuity of treatment, and the situation that the pandemic has brought about has strongly affected some of these factors, such as the doctor–patient relationship and access to clinics and pharmacies [13]. Many chronic patients have experienced a gap in their care, which may result in worse outcomes in the medium term. However, some preliminary observations on sales data suggest that restrictions and the lockdown implemented during the first pandemic wave (March–May 2020) resulted in excessive medication purchasing and stockpiling [14,15]. Moreover, some patients switched to mail-order pharmacies, and telehealth visits emerged as a viable alternative to support patients who may need a refill or a new prescription [16]. What is still unclear is whether the variation in drug prescribing habits has significantly affected chronic pharmacological regimens. Therefore, our study aimed to assess the impact of the COVID-19 pandemic on the continuity of medicines in patients already receiving chronic cardiovascular treatment.

## 2. Materials and Methods

### 2.1. Study Design, Population and Data Sources

In this population-based observational prospective cohort study, the target population included all non-institutionalized beneficiaries of the Italian National Health Service (NHS) aged 40 years or older, residents in Lombardy (an Italian region with 10.6 million inhabitants in 12 metropolitan areas) or the Local Health Authority (LHA) of Romagna (catchment area of approximately 1.1 million inhabitants), who were prescribed one of the chronic drugs of interest (Appendix A) between 1 July 2018 and 31 December 2018 (control cohort) or between 1 July 2019 and 31 December 2019 (study cohort). Three chronic drug treatments, lipid-lowering (LL), antihypertensive (AH), and antidiabetic (AD), were considered and analyzed separately. The drugs used to treat dyslipidaemia, diabetes, and hypertension, respectively, are covered by NHS reimbursement and therefore tracked in administrative databases. In Appendix A, the detailed list of Anatomical Therapeutic Chemical (ATC) codes is reported.

The healthcare administrative pharmaceutical databases of each Health Regional System were used to extract the treatment cohorts.

The study was carried out in conformity with the regulations on data management with the Italian law on privacy (Legislation Decree 196/2003 amended by Legislation Decree 101/2018). Data were anonymized prior to analysis at the regional statistical office, where each patient was assigned a unique identifier. This identifier does not allow others to trace the patient’s identity and other sensitive data. Anonymized regional administrative data can be used without specific written informed consent when patient information is collected for healthcare management and healthcare quality evaluation and improvement (according to art. 110 on medical and biomedical and epidemiological research, Legislation Decree 101/2018). 

The study was approved by the “Comitato Etico della Romagna” (C.E.ROM.) of the Emilia-Romagna Region, study protocol 9505/2020 on 14 December 2020, and carried out according to the Lombardy Region laws on the use of regional healthcare databases for research activities (D.g.r. XI/491, 2 August 2018; Decreto n. 16256, 12 November 2019), and in particular on COVID-19 disease (D.g.r XI/3019, 30 March 2020). 

### 2.2. Identification Algorithms and Inclusion Criteria

For each chronic drug treatment, two cohorts of prevalent cases were examined. The study cohort, one for each chronic therapy (study-LL, study-AH, study-AD), included all individuals with (1) at least one prescription of the chronic drug between 7 January 2019 and 31 December 2019, (2) alive as of 1 January 2020 (cohort entry), (3) aged ≥ 40 years, (4) resident in Lombardy or Romagna, and (5) with at least one prescription of the chronic drug during 2020. With the same inclusion criteria, the control cohort was extracted for each chronic therapy (control-LL, control-AH, control-AD) considering all the dates of the preceding year (at least one prescription between 7 January 2018 and 31 December 2018, and alive as of 1 January 2019).

The study and control cohorts of each chronic therapy were followed up to 12 months after cohort entry (1 January). Hospitalizations and all-cause deaths within 12 months of cohort entry were considered censored events.

### 2.3. Study Outcomes

The first outcome of this study was the number of pharmaceutical packages dispensed during the year in the study cohort and the control cohort; the second outcome was the adherence to drug prescription in the two cohorts, calculated with the proportion of days covered (PDC) method.

### 2.4. Measure of Adherence

To measure adherence to each chronic therapy, a supply diary was created for each patient-day by stringing together consecutive fills of each medication class being studied based on dispensing dates and reported days’ supply, which was estimated according to the total number of defined daily doses (DDD) (Appendix A). In the case of prescriptions dispensed during the recruitment periods with an estimated end date in the following year, we only considered the days covered after the 1st of January in the adherence calculation. For each chronic therapy (LL, AH, and AD), all drugs dispensed within a therapeutic class (ATC 4th level) were considered interchangeable: when dispensing occurred before the previous dispensing should have run out, utilization of the new medication was assumed to begin the day after the end of the old dispensing. 

The estimated adherence (PDC) was calculated as the ratio between the number of days during which patients had at least 1 of their prescribed medications of any class available to them and the number of days of observation [17].

### 2.5. Statistical Analysis

Demographic statistics of the study and control cohorts were summarized using frequencies and percentages, or mean and standard deviation, as appropriate.

The weekly trend of the mean pharmaceutical packages dispensed per 1000 inhabitants was analyzed in the study cohort and the control cohort, considering the uncensored subjects at the end of each week as the denominator.

Six sub-periods of analysis were also considered: weeks 1–9 (P1), weeks 10–18 (P2, full lockdown), weeks 19–26 (P3, restriction lighting), weeks 27–35 (P4), weeks 36–44 (P5), and weeks 45–52 (P6) (Appendix A). As 2020 was a leap year, the 31st of December 2020 was not considered in the analyses. The percentage variation of the mean pharmaceutical packages dispensed per 1000 inhabitants between the study cohort and control cohort was calculated in each sub-period, stratifying by sex, age group (40–64, 65–74, and ≥75 years), and metropolitan area.

Continuity of drug therapy in chronically treated patients is a multifactorial issue, influenced by several patient-, disease-, and system-related factors [18]. In the attempt to isolate the potential effect of the pandemic, we focused our attention on the most compliant subjects. Therefore, for the second outcome, individuals with a PDC ≥ 80% in the P1 period (January–February) were selected. In the next four-month period (P2 and P3), which approximately represents the first wave of the pandemic, the individual PDC was calculated, classifying the subjects according to adherence level: low-adherence (PDC <20%), middle-adherence (PDC 20–79%), and high-adherence (PDC ≥ 80%).

A descriptive comparison of sex, age, concomitant therapies, and comorbidities characteristics was examined between the subjects with low adherence to prescriptions in the four-month period (P2–P3) and those with high adherence to prescriptions in the same period. The χ2 test was used to compare categorical variables, and the t-test was used to compare group means.

For all tests, significance was set as *p* < 0.05. Statistical analyses were performed using the Statistical Analysis System Software (version 9.4; SAS Institute, Cary, NC, USA) and IBM Corp. Released 2017. IBM SPSS Statistics for Windows, Version 25.0. Armonk, NY, USA: IBM Corp.

## 3. Results

A total of 973,948 patients on LL (mean age [SD]: 71.1 years [10.6]; males: 52.2%), 421,231 on AD (mean age [SD]: 70.7 years [11.2]; males: 56.3%), and 2,311,093 on AH (mean age [SD]: 70.3 years [12.1]; males: 47.7%), were enrolled in the study period and compared with 948,164 control-LL (mean age [SD]: 70.9 years [10.6]; males: 52.2%), 411,747 control-AD (mean age [SD]: 70.5 years [11.1]; males: 56.1%), and 2,304,934 control-AH (mean age [SD]: 70.1 years [12.1]; males: 47.5%), respectively. 

Overall, there was a small decrease in the number of dispensed packages, comparing the entire year of 2020 with the previous one, less marked for LL (−2.22%) than other therapies (AD: −5.40%; AH: −5.01%). However, in all the cohorts, a slight increase was observed in P2 (LL: +4.52% [Figure 1 panel A]; AD: +2.72% [Figure 1 panel B]; AH: +1.09% [Figure 1 panel C]), with a sharp decrease in P3 and P6 (LL: −8.40% and −6.27% [Figure 1 panel A]; AD: −12.09% and −9.71% [Figure 1 panel B]; AH: −10.54 and −9.04% [Figure 1 panel C]). 

In the stratified analysis by sex, only small differences were observed between men and women in the one-year trend for each treatment (Appendix A). Consistent with the main analysis, for all the chronic therapies, we reported an increase in the number of dispensed packages registered in P2 (variations were slightly higher for men with AD and for women with LL and AH) and a sharp decrease in P3, both for females and males. Similar trends were observed considering age strata (Appendix A), with patients aged ≥ 65 years appearing more prone to stockpiling compared to younger patients. 

Geographic variability in trends is visible among metropolitan areas (Appendix A), in particular taking into account the different impacts that the pandemic had on them (those in dark blue represent areas with more COVID-19 cases recorded in 2020, while in light blue those with fewer cases). In the P2 period, we observed a higher increase in the number of dispensed packages in geographical areas less affected by the pandemic, while in all the other following two-month periods, the decrease in the number of dispensed packages compared to 2019 was confirmed for all three chronic therapies, regardless the pandemic impact.

When adherence was analyzed by calculating PDC, the impact of the COVID-19 pandemic on chronic cardiovascular treatments appeared negligible. In total, 533,414 of the patients showed high adherence to LL (PDC ≥ 80%) in P1 (with no difference in the proportion and patient characteristics between 2020 and 2019, Table 1), and 2.29% became poorly adherent (PDC < 20%) in the P2–P3 period in 2020 (vs. 1.98% in 2019). A similar increase in the proportion of patients who became poorly adherent was also observed for AH (P2–P3: 1.25% with poor adherence in 2020 vs. 0.93% in 2019), with the concomitant decrease of patients with optimal adherence (PDC ≥ 80%) from 89.15% in 2019 to 87.93% in 2020. For AD, the increase was restrained (P2–P3: 1.55% with poor adherence in 2020 vs. 1.37% in 2019) (Table 1). 

Comparing the subjects with low adherence to prescriptions in P2–P3 in terms of demographic and pharmacological characteristics, we found that patients becoming poorly adherent in P2–P3 had the same characteristics in the two years being observed (Table 2), regardless of the chronic therapy. The prevalence of subjects in polypharmacy or excessive polypharmacy significantly changed over the years; however, the magnitude of the differences was negligible, and the trend was not consistent among chronic therapies.

## 4. Discussion

Italy was one of the hardest hit countries during the COVID-19 pandemic [19,20]. 

Our analysis of the trend in dispensing volumes showed that, with the beginning of lockdown, patients already being treated for chronic cardiovascular diseases reacted by stockpiling drugs. When monthly trends were analyzed, an increase in dispensations occurred in March 2020, up to 15% for lipid-lowering agents. To some extent, such an attitude is understandable, considering that the early stages of the pandemic were characterized by a sense of precariousness, the indefiniteness of the future, and a strong concern for one’s health [21]. Other studies reported similar evidence in different countries. Clement et al. evaluated the continuity of care for six therapeutics classes less likely to be affected by the COVID-19 pandemic: hormonal contraception, immunosuppression, serotonin regulation, and drugs to address attention deficit hyperactivity disorder (ADHD) and psychoses. They found that more prescriptions were filled in March 2020 than in any prior month, followed by a significant drop in monthly dispensing [15]. Another study evaluating the English Prescribing Dataset from January 2014 to November 2020 showed that the prescription of numerous chronic medications was above the predicted values for March 2020 [21]. 

In the next months, dispensations in our Italian cohort probably decreased because patients drew from their stocks (in May 2020, dispensations were lower than in 2019 by 16–19%). Notwithstanding, our analysis showed that the impact of the COVID-19 pandemic on chronic cardiovascular therapies in already treated patients appears negligible in terms of overall dispensed packages as well as adherence of individual patients. When dispensations were considered, a 5% yearly decrease in antidiabetics and antihypertensives resulted in the worst finding, while lipid-lowering agents decreased to a smaller extent. As for adherence to therapy, patients with high adherence just before the pandemic (January–February 2020) maintained their adherence in the first wave period (March–June 2020): the number of patients who turned poorly adherent (PDC < 20%) was higher than in the same period during 2019, 3 out of 1000 for lipid-lowering and 2 out of 1000 for antidiabetics. In the case of antihypertensives, 1 out of 100 became medium adherent and 3 out of 1000 became low adherent. 

The impact of the pandemic has only been partially investigated in the literature. In the context of low- and middle-income countries, which rely heavily on pharmaceutical imports, the consequences of the unavailability of essential medicines have been described [22,23]. In other countries, as in Italy, the impact of the pandemic on chronic disease management seems to be essentially linked to lockdowns and restrictions on people’s movement and mobilization of health personnel to the frontline of the SARS-CoV-2 infection, with repercussions for patients with chronic diseases requiring revisits, follow-ups, check-ups, and prescription refills [24]. Nevertheless, in terms of the continuity of drug treatment, the impact of the pandemic appears to have been limited, and this evidence is also reported by other studies. Evaluating epilepsy care during the pandemic period, from March 2020 to May 2020, Muller et al. showed reduced care for newly diagnosed persons with epilepsy, while the adherence of persons with known epilepsy appeared to remain stable during lockdown in Germany [25]. In a study investigating medication adherence in patients with chronic obstructive pulmonary disease in Shanghai, China, during the COVID-19 pandemic, reported values were similar to those during regular times [26].

The minimal differences in terms of adherence to therapy found in Italy and other countries suggest that the tendency to stockpile drugs at the beginning of the lockdown period allowed sufficient coverage in the following months. On the other hand, the effectiveness of local and regional regulatory measures facilitating electronic prescriptions and promoting telemedicine and remote medical assistance should probably also be recognized [27]. Cross-country surveys [28,29] showed that, during the pandemic, electronic prescriptions were available in almost all European countries. In Italy, measures were taken at the beginning of the first wave of the pandemic to facilitate electronic prescribing, as well as electronic therapeutic plans [30,31]. The impact of the pandemic also fostered the use of telemedicine in the LHA of Romagna; specifically, some departments used remote assistance and digital tools to guarantee continuity of care to their patients whenever possible [32]. Our findings suggest an overall efficacy of Italian measures in limiting potential severe consequences of the pandemic on therapeutic continuity.

### Strengths and Limitations

In our study, we used secondary data, as they are routinely gathered at the individual level for administrative purposes and as a part of the healthcare system in Italy. The administrative databases themselves are an element of strength, as they collect all the reimbursed drugs dispensed to all citizens covered by the NHS. Administrative data collection, managed at a regional level, is nationally standardized, extremely accurate, and commonly used for drug utilization and pharmacoepidemiological research [33].

The choice of the method for calculating adherence sought to take into account the specificities of the evaluated treatments (in some cases, consisting of combination therapies) and the extension of coverage over time due to the accumulation of packages, which should be considered a strength of the study.

One of the main limitations of this study is that the assessment of adherence through PDC becomes less robust as the length of the observation period decreases. This limited our ability to stratify our sample to highlight subgroups of subjects on whom the impact of the pandemic was most dramatic and to study their characteristics. Another potential limitation of our study was that it did not fully elucidate patient attitudes to their health conditions, as only information regarding dispensed prescriptions could be captured, and adherence was assessed based on the assumption that the dispensing of a prescribed drug corresponded to the patient taking it. 

## 5. Conclusions

Our analysis provides some of the first published quantifiable impacts of the COVID-19 outbreak on primary care prescribing in Italy and further adds to existing evidence regarding the effect of the pandemic on the continuity of treatment for cardiovascular chronic health conditions. However, social and economic determinants could have influenced the continuity of use of chronic therapies during the pandemic, but the retrieval and global standardization of this information and relevant data linkage with large healthcare databases are still hard. Further studies, including the retrieval of data on social disadvantage, could allow us to define specific strata of the population associated with lack of coverage and drive healthcare policymakers to limit differences in healthcare service access. In the future, studies including data on disease worsening (hospitalization, laboratory data, etc.) could identify consequences of missing monitoring on health outcomes unrelated to medicine dispensation.

## Figures and Tables

**Figure 1 ijerph-19-12101-f001:**
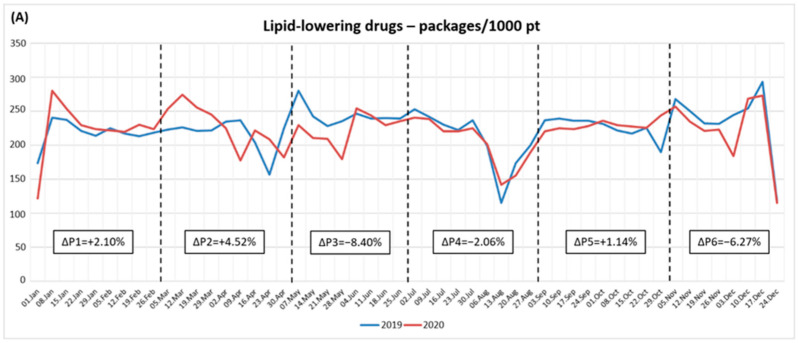
Fifty-two-week trends in chronic therapies dispensing in the population expressed as the number of dispensed packages per 1000 patients. (**A**) Lipid-lowering drugs. (**B**) Antidiabetic drugs. (**C**) Antihypertensive drugs.

**Table 1 ijerph-19-12101-t001:** Characteristics of subjects with high adherence (PDC ≥ 80%) at the end of the period P1, for each year and by the three chronic therapies.

	2019	2020	*p*-Value
**Lipid-lowering treatment**			
Number of subjects (% of the whole cohort)	533,414 (53.59)	555,817 (54.01)	
Male sex, %	54.25	54.23	0.789
Mean Age, y (SD)	70.83 (10.51)	71.02 (10.52)	<0.001
PDC classes (P2–P3)			<0.001
% Patients PDC < 20%	1.98	2.29	
% Patients PDC 20–79%	19.70	19.34	
% Patients PDC ≥ 80%	78.32	78.36	
**Antidiabetic treatment**			
Number of subjects (% of the whole cohort)	217,109 (50.62)	221,918 (50.33)	
Male sex, %	57.97	58.31	0.019
Mean Age, y (SD)	69.84 (10.74)	70.02 (10.76)	<0.001
PDC classes (P2–P3)			<0.001
% Patients PDC < 20%	1.37	1.55	
% Patients PDC 20–79%	13.88	13.73	
% Patients PDC ≥ 80%	84.23	84.27	
**Antihypertensive treatment**			
Number of subjects (% of the whole cohort)	1,541,649 (64.76)	1,539,530 (64.13)	
Male sex, %	64.76	64.13	<0.001
Mean Age, y (SD)	70.39 (11.84)	70.59 (11.85)	<0.001
PDC classes (P2–P3)			<0.001
% Patients PDC < 20%	0.93	1.25	
% Patients PDC 20–79%	9.46	10.40	
% Patients PDC ≥ 80%	89.15	87.93	

**Table 2 ijerph-19-12101-t002:** Characteristics of subjects with PDC < 20% at the end of the P2–P3 period for each year and by the three chronic therapies, among those with high adherence in P1.

Patients with PDC < 20%	2019	2020	*p*-Value
**Lipid-lowering treatment**			
Number of subjects (%)	10,537 (1.98)	12,740 (2.29)	
Male sex, %	48.94	48.18	0.252
Mean Age, y (SD)	70.01 (11.67)	69.90 (11.56)	0.460
Subjects aged ≥ 65 years, %	68.52	67.57	0.127
Polypharmacy [≥5 drugs], %	55.97	51.47	<0.001
Excessive polypharmacy [≥10 drugs], %	11.21	10.03	0.004
**Antidiabetic treatment**			
Number of subjects (%)	2971 (1.37)	3440 (1.55)	
Male sex, %	56.51	56.22	0.834
Mean Age, y (SD)	69.76 (12.23)	69.36 (11.95)	0.181
Subjects aged ≥65 years, %	66.85	65.49	0.265
Polypharmacy [≥5 drugs], %	53.99	56.92	0.020
Excessive polypharmacy [≥10 drugs], %	14.27	15.67	0.127
**Antihypertensive treatment**			
Number of subjects (%)	14,339 (0.93)	19,177 (1.25)	
Male sex, %	45.93	45.12	0.144
Mean Age, y (SD)	67.03 (13.16)	67.32 (12.46)	0.050
Subjects aged ≥ 65 years, %	55.65	56.90	0.022
Polypharmacy [≥5 drugs], %	30.84	26.61	<0.001
Excessive polypharmacy [≥10 drugs], %	4.27	3.78	0.024

## Data Availability

The pooled data that support the findings of this study are available from the corresponding author, M.C., upon reasonable request.

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
