# Peer review of "Impact of the COVID-19 Pandemic on the Therapeutic Continuity among Outpatients with Chronic Cardiovascular Therapies"

_ijerph, 2022, doi:10.3390/ijerph191912101_

Round 1

Reviewer 1 Report

I read with interest the paper by Casula et al. This is a study on pandemic impact on prescription and adherence pattern of chronic cardiovascular therapies (lipid-lowering, oral antidiabetic drugs, and antihypertensives). The study used administrative databases. 

I found the paper well written and I have only one main question: Authors analysed time series data after and before specific time points; why they did not perform a time-series interrupted analysis?

Other minor points:

1.      Line 68: I suggest to specify drugs of interest here as done in the "identification algorithms and inclusion criteria" paragraph

2.      Line 70: Is the study period 2020 or 2019?

3.      Lines 129-131. I suggest introduction figure showing study period to improve the understanding.

4.      Line 147: If I understand correctly, in the study cohort there are subjects using the drugs of interest in 2020 and with at least one prescription in 2019; while in the control cohort there are subjects using the drugs of interest in 2019 and with at least one prescription in 2018. This means that the authors included prevalent users, so part of the control sample is included in the study sample. Is this correct? If so, how did the authors handle the paired data?

5.      Lines 149-151: I cannot see the decrease of dispensations in the figure. Also, did authors test for differences between cohorts?

6.      Lines 199-200: Authors found that patients becoming poorly adherent in P2-P3 had the same characteristics in the two observed years. Did They test for  differences?

Author Response

I read with interest the paper by Casula et al. This is a study on pandemic impact on prescription and adherence pattern of chronic cardiovascular therapies (lipid-lowering, oral antidiabetic drugs, and antihypertensives). The study used administrative databases.

We would like to thank the Reviewer for the comment and we offer the following answers to her/his remarks.

I found the paper well written and I have only one main question: Authors analysed time series data after and before specific time points; why they did not perform a time-series interrupted analysis?

Thank you for your comment. The interrupted time series analysis is often used to quantify the impact of population-level health interventions on processes of care and population-level health outcomes, dividing time into “pre-intervention” and “post-intervention” periods. In our study, it is complex to identify a period of “intervention” (spread of COVID-19 pandemic), not only because we take into account more than one lockdown periods in Italy, but also because within each lockdown period COVID-19 pandemic has been a dynamic phenomenon. We are not sure if we can speak of an actual intervention, nor that the assumptions require by this analysis are fully validated. Moreover, the evaluation of differences in trends between years, and not within the same year, allowed to minimize the bias that might be introduced by seasonality.

Minor points:

  1. Line 68: I suggest to specify drugs of interest here as done in the "identification algorithms and inclusion criteria" paragraph

Thank you for your suggestion. We specified drugs of interest in the Study design, population and data sources paragraph.

  1. Line 70: Is the study period 2020 or 2019?

For each chronic therapy (LL, AH, and AD) we defined two cohorts: one CONTROL COHORT, including subjects who were prescribed one of the chronic drugs of interest between July 01, 2018 and December 31, 2018 (control period) and then followed throughout the year 2019, and one STUDY COHORT, including subjects who were prescribed one of the chronic drugs of interest between July 01, 2019 and December 31, 2019 (study period) and then followed throughout the year 2020. We better clarified the definition in the text.

  1. Lines 129-131. I suggest introduction figure showing study period to improve the understanding.

Thank you for your suggestion. We have now added a Supplementary Figure S2.

  1. Line 147: If I understand correctly, in the study cohort there are subjects using the drugs of interest in 2020 and with at least one prescription in 2019; while in the control cohort there are subjects using the drugs of interest in 2019 and with at least one prescription in 2018. This means that the authors included prevalent users, so part of the control sample is included in the study sample. Is this correct? If so, how did the authors handle the paired data?

Yes, it is correct: as we selected prevalent users, part of the control sample could be included in the study sample. However, we considered the two prevalent cohorts as independent samples because they did not include exactly the same subjects, therefore it is not possible to apply the test for paired data. More importantly, we were not interested in evaluating the different prescribing behaviours within subject, but the evaluation was focused on comparing the prescribing approach in two different years.

  1. Lines 149-151: I cannot see the decrease of dispensations in the figure. Also, did authors test for differences between cohorts?

Thank you for your comment. The sentence “Overall, there was a small decrease in the number of dispensed packages, comparing the entire year 2020 with the previous one, less marked for LL (-2.22%) than other therapies (AD: -5.40%; AH: -5.01%).” was referred to the difference between annual packages dispensed in 2020 (Jan-Dec) and annual packages dispensed in 2019 (Jan-Dec), as percentage on 2019 values. This is not reported in Figure 1, where we reported the same information by study periods instead. As we estimated just the difference between number of packages, statistical tests were not applicable.

  1. Lines 199-200: Authors found that patients becoming poorly adherent in P2-P3 had the same characteristics in the two observed years. Did They test for differences?

Thank you for your comment. We added in table 2 the test for differences and modified the text accordingly.

“Comparing the subjects with low adherence to prescriptions in P2-P3 in terms of demographic and pharmacological characteristics, we found that patients becoming poorly adherent in P2-P3 had the same characteristics in the two years being observed (Table 2), regardless of the chronic therapy. The prevalence of subjects in polypharmacy or in excessive polypharmacy significantly changed across years, however the magnitude of the differences was negligible and the trend was not consistent among chronic therapies.”

Reviewer 2 Report

I really enjoyed this manuscript and feel that the authors did an outstanding job describing a complicated cohort with enough detail to understand the process.  The description of the statistical methods included a clear definition of the outcomes and how they were collected.  The authors were judicious in listing what they felt were the strengths and limitations of the project, and the conclusions were supported by the results.

I have two very minor suggestions to increase clarity for the reader.

1. Lines 114-115

I don’t understand what this sentence is saying "The remains days covered by a prescription occurred in the six months preceding the cohort entry were considered at the beginning of the year".  It may just be a wrong word, but what does the “remains days” refer to?  

2. Figure 1:

A suggestion to make reading a bit easier would be to include on the figures the notion for time period (i.e., P1, P2, P3, etc.).  When reading the text describing the results, it required me at first to go back and forth to figure out what portion of the graph was representing the P1 and other time frames.  I realize they are consecutive, but having that information directly on the graphs would initially be very helpful to the reader.

Author Response

I really enjoyed this manuscript and feel that the authors did an outstanding job describing a complicated cohort with enough detail to understand the process.  The description of the statistical methods included a clear definition of the outcomes and how they were collected.  The authors were judicious in listing what they felt were the strengths and limitations of the project, and the conclusions were supported by the results.

We really appreciate the reviewer's positive feedback and we offer the following answers to his/her remarks.

Minor suggestions to increase clarity for the reader.

  1. Lines 114-115

I don’t understand what this sentence is saying "The remains days covered by a prescription occurred in the six months preceding the cohort entry were considered at the beginning of the year".  It may just be a wrong word, but what does the “remains days” refer to? 

Thank you for the comment. We have now detailed this approach in the text.

“In case of prescriptions dispensed during the recruitment periods with estimated end date in the following year, we only considered the days covered after January the 1st in the adherence calculation”

  1. Figure 1:

A suggestion to make reading a bit easier would be to include on the figures the notion for time period (i.e., P1, P2, P3, etc.).  When reading the text describing the results, it required me at first to go back and forth to figure out what portion of the graph was representing the P1 and other time frames.  I realize they are consecutive, but having that information directly on the graphs would initially be very helpful to the reader.

Thank you for your suggestion. The figure already includes the notion for the time period (the information is next to the DELTA symbol in subscript format), but it is probably written too small, so we have modified the figure to make it clearer. We have also added a Supplementary Figure S2 to improve the understanding of study periods.
